

# Measured particle water uptake enhanced by co-condensing vapours

Dawei Hu, David Topping, Gordon McFiggans*

School of Earth and Environmental Sciences, University of Manchester, UK.

Correspondence to: Gordon McFiggans (g.mcfiggans@manchester.ac.uk)

**Co-condensation of inorganic or organic vapours on growing droplets could significantly enhance both cloud condensation nucleus (CCN) and cloud droplet**

**number concentration, thereby influencing cloud albedo and climate. Until now, there has been no direct observational evidence of this process. We have measured the growth of inorganic salt particles exposed to both water and organic vapours at 291.15 K in the laboratory, showing that co-condensation of the organic vapours significantly enhances water uptake of aerosols. After exposure to water and propylene glycol vapours,**

**ammonium sulphate particles grew much more than any previously measured particles, inorganic or organic, at the same relative humidity. The maximum equivalent hygroscopicity parameter, $\kappa$, was observed to reach up to 2.64, very much higher than values ($0.1 < \kappa < 0.9$) measured for atmospheric particulate matter using conventional instrumentation, which may be blind to this effect. Under a continuously replenishing**

**organic vapour field, the particles never reached equilibrium owing to the presence of the involatile solute and were observed to continuously grow with increasing exposure time, in agreement with model simulations. Co-condensation of butylene glycol (which has similar volatility but, at $a_w = 0.9$, a higher $S_{org}$ than propylene glycol in our system) and tri-ethylene glycol (which has lower volatility and, at $a_w = 0.9$, lower $S_{org}$ than**

**propylene glycol in our system) vapours additionally measured in this study. The maximum equivalent hygroscopicity parameter, $\kappa$, reached as high as 8.48 for**



**ammonium sulphate particles exposed to water and tri-ethylene glycol vapours at 90% RH. This enhancement of particle water uptake through co-condensation of vapours constitutes the first direct measurement of this process, which may substantially**

**influence cloud droplet formation in the atmosphere. In addition, the model simulations for exposure to co-condensing butylene glycol and tri-ethylene glycol vapours with water show that there are factors other than $S_{org}$, which influence the co-condensation of SVOCs that are as yet not understood.**

## 1  Introduction


Clouds have a profound influence on weather and climate. According to the Intergovernmental Panel on Climate Change (IPCC), the impacts of aerosols on clouds are one of the largest uncertainties in estimates of global radiative forcing (Denman et al., 2007). Particle size, composition, mixing states and various derived properties such as

hygroscopicity are the main factors to determine if particles can act as a cloud condensation nucleus (CCN) and then form the cloud droplets under atmospheric water saturation ratio (McFiggans et al., 2006;Dusek et al., 2006;Topping and McFiggans, 2012).

The formation of cloud droplets by the condensation of water vapours on particles can be predicted by traditional Köhler theory (Köhler, 1936). The theory, in its original unmodified

form, was designed for particles comprising involatile components in the presence of a single supersaturated vapour. In addition to semi-volatile inorganic gases such as ammonia and nitric acid, there are many organic compounds of varying volatility (McFiggans et al., 2010) in the atmosphere which, if they were to influence water uptake, would substantially affect cloud properties (Topping et al., 2013). Kulmala et al. (1993) first suggested that co-

condensation of atmospheric $HNO_3$ could alter the activation and growth of cloud condensation nucleus (CCN) significantly and this was extended to more comprehensively



consider more complex inorganic systems including $NH_3$ (Hegg, 2000;Xue and Feingold,

2004;Laaksonen et al., 1998;Romakkaniemi et al., 2005). Recently, co-condensation of

organic vapours to the growing droplets was also suggested to substantially enhance both

CCN and cloud droplets number concentration, thereby influencing cloud albedo. The

cooling tendency from a net influence on cloud albedo associated with the predicted

enhancement in droplet number was estimated to be of the order of 1 $Wm^{-2}$. However, all of

these studies were solely theoretical. Current weather, climate, air quality or Earth System

models do not include this process. There has been no previous direct measurement evidence

for this process in either inorganic or organic systems and existing instrumentation that is

used to inform and challenge models has not been designed to be sensitive to this effect.

To directly probe co-condensation, we consider the case of a particle comprising a single

involatile water soluble solute. Exposure to an environment with constant saturation ratio of

water and semi-volatile organic vapours ($S_w$ = relative humidity, RH, and $S_{org}$ respectively)

should lead to condensation of both vapours towards an equilibrium particle composition and

size. However, the ability of the particle to reach such a state will be inhibited by the

presence of the original involatile solute. This will result in the condensed phase activity of

the involatile component persisting above the vapour phase saturation ratio and condensed

phase activities of the semi-volatile components below their saturation ratios. Water and

organic vapour will therefore continue to condense and the droplets will continuously grow

by the same process that has been hypothesised to lead to clouds of stable supermicron

droplets below water supersaturation (Kulmala et al., 1997).

In this study, ammonium sulphate, a representative atmospheric electrolyte, was chosen as the

involatile solute. Propylene glycol (PG), butylene glycol (BG) and tri-ethylene glycol (TEG)

were selected as the semi-volatile organic compounds since they are completely miscible

with water across their concentration ranges, thereby reducing the likelihood of abrupt



changes in ideality associated with phase transitions. These compounds were deliberately chosen to avoid complicating factors such as immiscibility and solubility limitation that would confound straightforward quantitative interpretation. Such cases may show less

dramatic influences of co-condensation, nevertheless they would still only enhance water uptake, never reduce it below the case where there was no co-condensation. It is not the intention to fully explore all possible atmospheric behaviours, which must remain the scope of future work.

## 2   Methods

### 2.1 Experimental setup

### 2.1.1 Sub-saturation ratio hygroscopic growth measurement

Figure 1 shows the experimental setup for investigating the influence of co-condensation of organic vapours on the water uptake of $(NH_4)_2SO_4$ particles. The configuration is based on a modification of the self-made Hygroscopicity Tandem Differential Mobility Analyser

(HTDMA) system at the University of Manchester (Good et al., 2010). Briefly, polydisperse $(NH_4)_2SO_4$ particles produced by an atomizer are dehydrated to ~10% RH through a Nafion drier and then neutralized with a $^{90}Sr$ diffusion charger before size-selected ($D_0$) by the first DMA (DMA-1, Brechtel Manufacturing Inc., USA). Quasi-monodisperse particles exiting DMA-1 are subsequently exposed to water and organic vapors in a glass reactor containing

organic-water solution with water activity at 0.9. The particles are thus grown in a continually replenished RH of 90% and a certain $S_{org}$ (depends on non-ideality). The particle diameter ($D_p$) is measured using a second DMA (DMA-2, Brechtel Manufacturing Inc., USA) before detection using a condensation particle counter (CPC, Model 3786, TSI Inc., USA). The hygroscopic growth factor (GF), defined as the ratio of the processed particle diameter ($D_p$)

over the initial dry particle diameter ($D_0$), was obtained through the inversion of scanning



DMA-2 data by the TDMAInv software (Gysel et al., 2009). The pure component properties of the semi-volatile PG, BG and TEG used in their aqueous solutions as the working fluids in our experiments are summarised in Table 1.

During the experiment, both DMAs and glass reactor were temperature controlled and held at 18 °C in a thermostatic box with a temperature fluctuation smaller than 0.2 K. RH, temperature and flows in the instrument were monitored at several locations and the RH and temperature measured at the sample outlet of the DMA-2 were used for subsequent calculation. The dew point sensor was found to be unsuitable for this study since the co-condensation of propylene glycol and water vapour on chilled mirror surface led to overestimation of RH. All RH sensors were capacitance sensors, and were found not to be influenced by the organic vapours in our experiment. To investigate the exposure time effects on the water uptake of $(NH_4)_2SO_4$ particles, 0.5, 2 and 4 m long glass reactors (I.D. = 2.4 cm) with 50, 200 and 400 ml prepared organic-water solution were studied separately, the corresponding residence time of particles in each glass reactor being 23.5, 94 and 188 s, respectively. Based on the sample flow rate (0.45 L/min) and glass cell dimensions, the calculated Reynolds number is less than 100, enabling the assumption of laminar flow.

### 2.1.2 Droplet activation measurement above water saturation

A Continuous Flow Streamwise Thermal Gradient Diffusion Chamber Cloud Condensation Nucleus counter (CFSTGDC CCNc, DMT) was used to attempt to measure the droplet activation of $(NH_4)_2SO_4$ particles in both water and organic vapours. The reactor outflow containing the grown $(NH_4)_2SO_4$ particles was split between the CCNc and a CPC (3775, TSI) instead of DMA2 in Figure 1. By stepping through different dry sizes in DMA1, the activation diameter ($D_{50}$) of $(NH_4)_2SO_4$ particles at a given water supersaturation (SS) was determined as the diameter at which 50% of the particle number concentration measured by the CPC were measured to be activated in the CCNc.



## 2.2 Instrument calibration and characterization

### 2.2.1 Calibration and certification of HTDMA system

Before the experiment, DMA-1 was calibrated with certified polystyrene latex spheres (PSLs). The offset in the size measurement of the two DMAs was determined with dry

$(NH_4)_2SO_4$ particles by using an empty glass reactor in the system to maintain the dry environment of DMA-2. In addition, to verify the performance of the modified H-TDMA system, GF of $(NH_4)_2SO_4$ particles were measured after passing through the headspace in the glass reactor containing NaCl-water solution with the water activity at 0.9. Since NaCl is involatile, the growth of $(NH_4)_2SO_4$ particles resulted solely through the uptake of water

vapour. As shown in Figure 2, for the 0.5 m glass cell with 50 ml NaCl-water solution (approximate average residence time = 23.5 s), the measured RH in DMA-2 sample line is 90 ± 0.3%, exactly corresponding to the equilibrium RH (90%) with the prepared NaCl-$H_2O$ solution. For a 0.25 m glass cell with 25 ml NaCl-water solution (residence time = 11.7 s), the measured RH was found to vary between 81 and 83%, lower than the expected value of

90%. This demonstrates that a 0.5 m glass cell is sufficiently long to allow dried $(NH_4)_2SO_4$ particles to reach the equilibrium RH (90% in this study) with the prepared solution whilst a 0.25 m reactor is not. Thus, the minimum length of the glass reactor used in this study is 0.5 m, sufficient to expose the particles to their equilibrium RH of 90%. For both the 0.25 and 0.5 cm glass reactor, the measured GF of $(NH_4)_2SO_4$ particles agree well with the Aerosol

Diameter Dependent Equilibrium Model (ADDEM) calculations (http://umansysprop.seaes.manchester.ac.uk/) at the corresponding RH, indicating the good performance of the modified H-TDMA system. Similarly good agreement is achieved in the longer reactors.



### 2.2.2 Calibration of CCN counter


Before the experiment, the CCN counter was calibrated with $(NH_4)_2SO_4$ particles. Briefly, $(NH_4)_2SO_4$ particles were generated in the same manner as for the HTDMA calibrations. Nafion dried $(NH_4)_2SO_4$ particles were neutralized using a [90]Sr diffusion charger, and size selected by a DMA before splitting the flow between the CCNc and a CPC (3775, TSI). The

DMA voltage was stepped to select each size and kept constant for 20 s, using data in last 10 s to calculate the ratio of $N_{CCN} / N_{CPC}$. The activation diameter ($D_{50}$) of $(NH_4)_2SO_4$ particles at a given water supersaturation (SS) was determined when the ratio of $N_{CCN} / N_{CPC} = 0.5$. The theoretical $SS_{critical}$ corresponding to the measured $D_{50}$ from the ADDEM model was used to calibrate the SS in the CCNc.

**2.2.3 The effect of the time for the system to reach the steady state**

Before commencing each experiment, the system was required to have reached steady state, such that vapour losses to, and outgassing from, the tubing, reactor and inside of the DMA-2 led to a stable measurement of GF. Figure 3 illustrates the effect of the time to reach the steady state of the system. For 0.5 m glass reactor with 50 ml aqueous PG solution, after the

system has continuously run for 70 hours, the measured GF of $(NH_4)_2SO_4$ particles after passing through the glass reactor becomes stable within the experimental uncertainty, i.e. the system reached the steady state. Similar behaviour was observed and duplicated for all reactor lengths and the particle GFs in this study are only reported after the system has reached steady state for all reactor lengths. This extended stabilisation period is the most

challenging practical limitations of the system.

**2.3 Model simulation**

The Aerosol-Cloud-Precipitation Interactions Model (ACPIM) was used to simulate the growth of monodisperse $(NH_4)_2SO_4$ particles with water and organic vapours using the same





numerical basis described in Topping et al. (2013). ACPIM is a numerical model of aerosol,

cloud and precipitation interactions that simulates the growth of particles of defined arbitrary

size or composition as they compete with available vapours to act as CCN in cloud droplet

formation or ice nucleus (IN) in ice crystal formation under evolving ambient environmental

conditions. In its usual configuration, ACPIM solves 4 coupled ordinary differential

equations for the water and organic vapour mass mixing ratio, pressure, temperature and

height of an air parcel rising through the moist atmosphere in hydrostatic balance. The model

was simplified to run at constant pressure and laboratory temperature, to treat an initially

monodisperse particle population and only one condensable organic vapour in addition to

water. The mass of water and organic compounds condensing to the particles by virtue of the

difference of vapour pressure between the particle surface and the surrounding air was

followed in the simulation. The dried sample flow does not instantaneously reach the water

and organic vapour saturation ratios on entry to the reactor but mixing ratios of both will

increase towards the equilibrium value with exposure time. The glass reactor walls compete

with the growing particles for both components, but the vapours are continually replenished

from the solution bath.

**3 Results and Discussion**

**3.1 Direct measurements of particle growth in co-condensing vapours**

**3.1.1 Hygroscopic growth measurements below water saturation**

Panel (a) in Figure 4 shows the measured and modelled growth of monodisperse $(NH_4)_2SO_4$

particles in the presence of water and PG vapours and panel (b) illustrates the difference in

the growth of $(NH_4)_2SO_4$ particles when exposed only to water vapour or when exposed to

the mixture of water and organic vapours. In the absence of organic vapours, $(NH_4)_2SO_4$

particles remain solid with increasing RH until the deliquescence RH (DRH) is reached, at



which point there is a distinct and abrupt increase in diameter as the particles undergo a solid to liquid phase transition. Further increase of the RH leads to additional water condensation on the salt solution and the particle increases in size. When the RH reaches equilibrium, the droplet size remains constant and is not influenced by the exposure time. When the $(NH_4)_2SO_4$ particles are exposed to water and organic vapours, both condense and the particle increases in size more rapidly than in the absence of organic vapours. Condensation depends on the concentration difference between the ambient air and the surface of droplet. Each semi-volatile compound (including water) tends to simultaneously condense from the vapour phase towards particles according to its prevailing saturation ratio. The sum of the vapour saturation ratios in the reactor headspace tends towards unity above the aqueous organic solution. The sum of the activities in the particles will always be a finite amount below unity owing to the presence of the involatile $(NH_4)_2SO_4$. This leads to the continuous existence of a driver towards condensational growth of the particles, such that the droplets cannot achieve equilibrium with surrounding organic vapour and water activities in the time allowed in the continuously replenishing vapour field. Owing to the presence of $(NH_4)_2SO_4$ solute in the droplet solution, the droplet will continuously grow throughout their period of exposure to the vapours.

Figure 4 shows that the measured GF of $(NH_4)_2SO_4$ particles increased significantly more than at the same RH where no condensable organic vapours are present (dotted line). After exposure to propylene glycol and water vapour for 188 s, dry 150 nm $(NH_4)_2SO_4$ particles increase in diameter to 422 nm (GF = 2.81), 160 nm greater than when exposed to water vapour only. This clearly shows that co-condensation of semi-volatile organic vapours greatly increases the water uptake of aerosol droplets, to such an extent that the measured growth factor at 90% RH is greater than has been measured even for the most hygroscopic inorganic salts in the absence of organic vapours. In addition, the GF of monodisperse $(NH_4)_2SO_4$



particles was observed to increase with length of reactor (and hence exposure time at a given

flow rate). For example, 150 nm dry particles were measured at 314, 362 and 422 nm with

reactor lengths of 0.5, 2 and 4 m (and exposure times of 23.5, 94 and 188 s) respectively.

These GF values would be equivalent to values of the hygroscopicity parameter, $\kappa$, of 1.00,

1.46 and 2.64 if the particles were considered to comprise involatile solute. Importantly, this

shows that the equivalent $\kappa$–value is dependent on the exposure time and amount of vapour

so the hygroscopicity in the presence of condensable vapours cannot be defined for a particle

if only the composition of the condensed material is known.

Co-condensation of SVOCs depends directly on RH and $S_{org}$. This is the ratio of the ambient

partial pressure to the vapour pressure under the ambient conditions ($p/p_0$). Increasing the

concentration difference between the ambient air and the surface of droplet will enhance co-

condensation. As shown in table 1, BG ($\log_{10}(C^*) = 5.64$) has very similar volatility to PG

($\log_{10}(C^*) = 5.71$), but exhibits higher activity, $a_{BG}$ (0.164) than $a_{PG}$ (0.1) at the same water

activity, $a_w$, of 0.9. According to our theoretical understanding, replacing the aqueous PG

solution with BG solution in our system should increase the difference of organic

concentration between the ambient air and the surface of droplet. This should, in turn enhance

the co-condensation of water and organic vapours to the particle phase, increasing the droplet

size by condensation. Contrary to this expectation, as shown in table 2, the GF and equivalent

$\kappa$–value of $(NH_4)_2SO_4$ particles exposed to the headspace above the aqueous BG solution in

the 2 m reactor were smaller than those exposed to PG and water vapours (though clearly still

higher than those exposed to water vapour alone). For 100 nm $(NH_4)_2SO_4$ particles, the GF

and equivalent $\kappa$–value were observed at 1.98 and 0.83 in former, lower than the

corresponding values of 2.19 and 1.11 in latter. Possible explanations will be discussed in the

section 3.2.2. In contrast to BG, TEG has a lower vapour pressure (corresponding to a

saturation concentration of $\log_{10}(C^*) = 3.27$) than PG and lower activity $a_{TEG}$ (0.026) than $a_{PG}$





(0.1) in aqueous solution of the same water activity, $a_w$, of 0.9. However, the GF and equivalent κ–value of $(NH_4)_2SO_4$ particles exposed into the 2 and 4 m glass reactor with the aqueous TEG solution were observed to be much greater than that with aqueous PG. As shown in Table 2 and the open circles (measured GF) in the top panel of Figure 5, the GF (and the corresponding calculated equivalent κ–value) of dry 75 nm particles were measured at 3.09 (3.39) and 4.19 (8.48) with reactor lengths of 2 and 4 m (and exposure times of 94 and 188 s) respectively, much larger than the corresponding values of 2.11 (1.02) and 2.56 (2.09) for the PG experiment.

### 3.1.2 Insensitivity of CCN counter to co-condensation of organic vapours

The CCN behaviour of $(NH_4)_2SO_4$ particles after exposure to PG ($a_{PG} = 0.1$) and water ($a_w = 0.9$) solution in the glass reactor was referenced to measurements made using NaCl aqueous solution with $a_w = 0.9$. As shown in Figure 6 (a), no clear difference of kappa and $D_{50}$ was observed and the kappa difference and $D_{50}$ difference was less than 0.05 and 3 nm, respectively. This results from evaporation of the organic vapour in the heated column whilst the water vapour from the wetted walls condenses onto the activating droplet. The operating principle of the continuous flow diffusion chamber type of CCN counter is to create a supersaturation down the instrument centre line through the slower diffusion of heat than of water vapour from the heated and wetted walls. Simultaneously in our experiment the saturation ratio ($p/p_0$) of PG will decrease on moving down the centre line of the column, since its saturation vapour pressure ($p_0$) increases with temperature. This would clearly favour evaporation rather than co-condensation of PG vapour. It is possible that there is an indication that the difference of kappa and $D_{50}$ may have been significant at the lowest setpoint SS only. Such behaviour would be consistent with the organic vapour being evaporated least with the lowest temperature difference used to create the low SS. Figure 6 (b) presents the temperature difference and the corresponding calculated ratio of $S_{org}$ between the



outlet and inlet of the CCNc under different SS. Our results show that the CCNc is insensitive to the co-condensation of organic vapours once the $S_{org}$ (outlet)/$S_{org}$ (inlet) decreased to
around 0.69.

This result demonstrate that instruments conventionally used to measure particle water uptake will be largely insensitive to, or be unable to quantitatively access, the co-condensation effect. Whilst humidity is controlled in such instruments, initial drying of the sample stream or heating within an instrument will likely suppress the saturation ratio of organic components
by decreasing the organic component mixing ratio or raising the saturation vapour pressure respectively.

### 3.2. Numerical model interpretation of co-condensing particle growth

To quantitatively understand co-condensation of organic vapours to the particle droplets, we attempted to simulate the growth of monodisperse $(NH_4)_2SO_4$ particles in water and organic
vapours using the Aerosol-Cloud-Precipitation Interactions Model (ACPIM)(Topping et al., 2013).

### 3.2.1 Monodisperse $(NH_4)_2SO_4$ particle growth in water plus propylene glycol (PG) vapours

In the ACPIM simulations, the RH (Figure 4(a), red line) and $S_{org}$ (green line) increase with
the residence time after the dry $(NH_4)_2SO_4$ particles enters into the glass reactor. The RH profile is constrained by the experimental data which shows that particles reach their equilibrium size at 90% RH within the 0.5 m reactor filled with NaCl-water solution. The $S_{org}$ profile was used to optimise the fit between simulated and measured GFs at each reactor length. As shown in Figure 4(a), the RH rapidly reaches equilibrium (~ 20 s) in the glass
reactor, while $S_{org}$ needs longer (~ 700 s) (corresponding to 15 m glass reactor). This is to be expected, since water is more volatile than PG, and water vapour lost to the walls will be



more rapidly replenished from the solution than the PG vapours. The best fit simulation indicates that the steady state between wall loss and replenishment from the solution bath does not allow the system to reach its equilibrium $S_{org}$ in the maximum available exposure

time in our experimental configuration. This increase in $S_{org}$ might therefore be expected to lead to an increasing GF with the residence time as we observe. The $(NH_4)_2SO_4$ present in solution would effectively preclude equilibration (at least delaying it until the inorganic mole fraction was negligible) and the concentration difference between the surrounding air and the surface of droplet will drive water and organic vapours continuously to condense to the

droplets. This is illustrated by the continuous increase in the simulated GF beyond the point at which the vapour reaches its equilibrium $S_{org}$ (residence time > 700 s). The GF of larger particles are both measured and simulated to be larger than small ones, reflecting the Kelvin term size dependence of the equilibrium water and propylene glycol content of the particles.

### 3.2.2 Monodisperse $(NH_4)_2SO_4$ particles growth in water plus butylene glycol (BG) and

**water plus tri-ethylene glycol (TEG) vapours**

The ACPIM simulated GF with residence time for $(NH_4)_2SO_4$ particles in water plus BG or TEG vapour are presented in Figure 7 and Figure 5. For both simulations, the RH and $S_{org}$ profile used the same exponential function as the previous simulation for PG, but the final $S_{org}$ was changed to 0.164 and 0.026 (see table 1), the corresponding solute activities of BG

and TEG respectively with the water activity, $a_w$, of 0.9. Since evaporation rate is controlled by the component vapour pressure (and since BG has comparable volatility to PG), using the same exponential function for RH and $S_{org}$ should be reasonable. As shown in Figure 7, the simulated GF using aqueous BG (solid black line) was greater than that using the aqueous PG solution (blue dash line in Figure 4). This results from the higher $S_{org}$ of BG, producing the

larger difference in organic concentration between the ambient air and the surface of droplet, enhancing the co-condensation of water and organic vapours, leading to a larger droplet. This



contrasts with a measured GF (2.14) of 150 nm $(NH_4)_2SO_4$ after transit through the 2 m glass

reactor containing aqueous butylene glycol solution (much lower than the predicted value of

2.65 and lower even than the 2.41 measured for the aqueous PG solution). This indicates that

some as yet unknown factor besides $S_{org}$ can substantially influence the co-condensation of

SVOCs. Figure 7 shows that good agreement can be achieved between the measured and

simulated (black dash line) GF by reducing the mass accommodation coefficient of BG from

1.00 to 0.02 in the model, though there is no reason to expect such a mass transfer limitation

in any of the systems investigated.

Using aqueous TEG as the working fluid, the simulated GF (dash line in Figure 5) for 2 and 4

m glass reactors was much lower than the measured value. This was the case even when

using the same exponential function for RH and $S_{org}$ as for the PG simulation. This should

lead to an overestimated GF since the volatility of TEG was 275 times lower than PG. A

correspondingly slower build-up of the TEG vapour concentration (and hence $S_{org}$) from the

reduced evaporation rate should result. Moreover, the simulated GF of larger particles is

larger than small ones (as with BG and PG), but the measured GF shows the largest value for

75 nm dry $(NH_4)_2SO_4$ particles, followed by 100, 50 and 150 nm. This is clearly inconsistent

with any of the physical representations in the model which predict greater growth for the

larger particles owing to the Kelvin effect and the controlling influences on the co-

condensation of SVOCs are not fully understood.

### 3.2.3 Co-condensation of Semi-VOCs with wide range of volatility

Whilst the volatility and concentration of each of the organic components (PG, BG and TEG)

used in this study is higher than most of the semi-VOCs in the atmosphere, their saturation

ratio is within a reasonable range of those of a wide range of condensable organic

components. The absolute magnitude of co-condensation depends on the saturation ratio, $S_{org}$

$(p/p_0)$, not the absolute values of the vapour pressure (p, reflected by the absolute





concentration) nor the saturation vapour pressure ($p_0$, representing the component volatility). By analogy, the hygroscopic growth of particles depends only on RH ($p/p_0$), not the absolute humidity (or water concentration) in the atmosphere. The magnitude of the co-condensation

effect will be identical if the same $S_{org}$ ($p/p_0$) was sustained for organic compounds of different vapour pressures, of low or high volatility.

To illustrate this, the GF profiles with residence time for different volatility organics ($\log_{10}(C^*)$ values= 5.71, 5, 4, 3, 2, 1, 0) were simulated using ACPIM in Figure 8. In each simulation in Figure 8(a), the RH and $S_{org}$ profile was kept the same (i.e. there was no

contributing effect on the kinetics of evaporation on the build-up of the gaseous saturation ratio). When $\log_{10}(C^*)$ is decreased from 5.71 to 4, (i.e., both volatility ($p_0$) and concentration (p) decrease by a factor of around 51), no difference was observed for GF profile, illustrating how $S_{org}$ is the controlling factor in the absolute magnitude of co-condensation, not the volatility or absolute concentration. Below $\log_{10}(C^*)$ <4, the growth is slowed for

components with decreasing volatility and the slope of the GF curve decreases with $\log_{10}(C^*)$ owing to reduced number of collisions, i.e., longer residence time of lower volatility organic compounds was needed to attain the same final GF. Figure 8(b) illustrates that a more rapid growth can be re-established by an increase in $S_{org}$ from 0.1 to 0.5 in the model for relatively low volatility organics ($\log_{10}(C^*)$=1) with low concentration (~ 0.9 ppb). It can be reasonably

expected that modest concentrations of organic components can be maintained by various mechanisms in the atmosphere (strong direct sources or in situ oxidative sources, for example) and co-condensation effects may be expected to play an important role in the real atmosphere.

**3.3 Uncertainties in the simulated experimental conditions – the influence of axial and radial diffusion effects on the magnitude of co-condensation**

In this study, the bulk velocity (V) was used to calculate the residence time of particles in the glass reactor, and was subsequently used in ACPIM simulation. In reality, owing to a non-



uniform radial velocity profile, particles close to the centerline (wall) in the glass reactor will have a larger (smaller) velocity than V, giving these particles a shorter (longer) residence time in the glass reactor, thus achieving a smaller (larger) GF by the condensation of organics and water vapour. To illustrate the possible magnitude of any effect of axial flow heterogeneity, the GF profile with the varying residence time was simulated by ACPIM and the yellow hatched area is bounded by the GF of particles with half or twice the nominal plug flow residence time (Figure 9 (a)), representing the variation in the axial flow velocity.

Organic vapours in the reactor headspace are diluted by the sample air and scavenged by the co-condensational growth of $(NH_4)_2SO_4$ particles. The growth process proceeds down a stronger concentration gradient at the beginning and becomes weaker with the residence time due to the decrease of vapor pressure difference between the particle surface and the surrounding air. It can be seen from the simulated $S_{org}$ profile in Figure 4, Figure 8 and Figure 9 that the organic vapour builds more slowly during the first few hundreds second early growth period, leading to a radial gradient in $S_{org}$ during this period, i.e., the $S_{org}$ close to the organic solution surface will more rapidly reach equilibrium. Particles passing through different radial positions will experience different $S_{org}$ profile. To examine the possible influence of such radial heterogeneity, two GF profile evolution extremes with residence time were simulated by ACPIM for the difference $S_{org}$ profile: one reaching equilibrium twice as rapidly as in Figure 4, the other being half as rapid. The yellow shading area shows influence of radial diffusion on the GF profile in Figure 9(b).

It can be seen that, whilst both axial and radial gradients and heterogeneities will exist within our experiments, the generality of our results and conclusions remain largely unaffected.








## 4. Implications for the atmosphere

Our experimental configuration requires that we use a relatively volatile organic compound to maintain its saturation ratio in the vapour phase through evaporation from the solution bath. In the atmosphere, such a limitation will not apply and the saturation ratio may be

continuously replenished through surface fluxes or via chemical reaction. A supersaturation is not required and the saturation ratio need only be maintained above the activity in the growing particle. Moreover, in the atmosphere, many thousands of semi-VOCs contribute to the 30-70% organic mass of the submicron aerosol particles (Jimenez et al., 2009;Hallquist et al., 2009) and all can co-condense. This huge diversity of organic components will exhibit a

wide range of volatilities and solubilities, nevertheless the interaction of condensing inorganic or organic components will only ever be to increase a particle's ability to act as a CCN under any reasonable assumptions. In the section 3.2.3, it is shown that co-condensation of organic components across a wide range of volatility of relevance to the real atmosphere can play a role comparable to that reported in our measurements. Non-ideality and phase

separation are complications that will occur in ambient mixtures and must be investigated in order to fully resolve the implications in the atmosphere. However, the exceptional complexity and huge abundance of organic vapours, the fact that they can only enhance water uptake by the measured effect and the fact that it is an individual component's saturation ratio and not absolute concentration that leads to this effect all mean that the effect of co-

condensing organic components cannot be ignored. ACPIM was previously used to show that cloud droplet number in the atmosphere could be enhanced by several tens of percent by co-condensation of organic vapours, with a potential to substantially increase the cooling tendency from a net influence on global albedo (Topping et al., 2013).

Co-condensation of semi-volatile vapours with water during growth in a moist atmosphere

and in subsequent cloud droplet formation will substantially challenge conventional



instrumentation employing diffusion drying or using thermal gradient to probe CCN properties. This current study has observed for the first time that co-condensation of organic vapours can significantly promote water uptake of aerosol particles, a process which, in the atmosphere, will significantly change particle activation as CCN, cloud droplet growth and subsequent influence on indirect radiative forcing.

## 5. Conclusions

Co-condensation of water and organic vapours to $(NH_4)_2SO_4$ particles was directly observed in the laboratory, in an attempt to evaluate the general construct of our theoretical framework. This first study aimed to examine the phenomenon in the most straightforward way, carefully designing the system to avoid complex and largely unquantifiable confounding process level phenomena such as immiscibility and limited solubility. From our study, we can conclude that:

(1) Co-condensation of the organic vapours significantly enhances water uptake of aerosols. In this study, the maximum equivalent hygroscopicity parameter, $\kappa$, was observed to reach up to 8.48, very much higher than values ($0.1<\kappa<0.9$) measured for atmospheric particulate matter using conventional instrumentation.

(2) Instruments conventionally used to measure particle water uptake will be largely insensitive to, or be unable to quantitatively access, the co-condensation effect. Whilst humidity is controlled in such instruments, initial drying of the sample stream or heating within an instrument will suppress the saturation ratio of organic components by decreasing the organic component mixing ratio or raising the saturation vapour pressure respectively.

(3) Residence times within such instruments would be too short for them to be sensitive to co-condensation at ambient concentrations of organic components even if they were retained in the instrument.

(4) ACPIM simulation can be tuned to readily reproduce the absolute values of effective hygroscopicity for some systems, but there are factors not considered by the model that play an important role in co-condensation of SVOCs.

Our work should serve as the basis for further investigation, providing the first experimental
evidence in this simple system with co-condensation to a single liquid phase. We will aim to include less water-soluble organics in the future, representatively quantifying the impact of Liquid-Liquid-Equilibria (LLE) will need to account for a representative spectrum of volatility and solubility such that any result is not sensitive to one particular compound or group of compounds. A possible need to account for the geometry of distinct phases resulting
from LLE in cloud activation has been suggested in some studies.

**Acknowledgements.** This work was supported through the UK Natural Environment Research Council (NERC) in the CCN-Vol project (grant ref: NE/L007827/1). The infrastructures used within CCN-Vol receive funding from the Horizon 2020 research and
innovation programme through the EUROCHAMP-2020 Infrastructure Activity under grant agreement no. 730997.

**Author contributions.** G.M. and D.H. designed research; D.H. performed experiments; D.T. developed the model code; G.M., D.T and D.H. designed the simulations; D.H. carried out data analysis; D. H. performed model simulation; D.H., G. M. and D.T. co-wrote the paper.

**Data availability.** Raw data is archived at the University of Manchester and is available on request.

**Competing financial interests**

The authors declare no competing financial interests.





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



545                                                        **Tables**

550                Table 1 Parameters of Propylene Glycol, Butylene Glycol and Tri-ethylene Glycol

| Organics | Molecular formula | Solubility in water | Activity with aw=0.9 | $Log_{10}$ (C*) @25°C (µg/m$^3$) | ΔH (Vap) (KJ/mole) |
|---|---|---|---|---|---|
| Propylene Glycol | $C_3H_8O_2$ | Miscible | 0.1 | 5.71 | 61.30 |
| Butylene Glycol | $C_4H_{10}O_2$ | Miscible | 0.164 | 5.64 | 59.91 |
| Tri-ethylene Glycol | $C_6H_{14}O_4$ | Miscible | 0.026 | 3.27 | 90.67 |


Table 2 Growth factor and the equivalent kappa of $(NH_4)_2SO_4$ particles exposed in Propylene Glycol/water

vapours, Butylene Glycol/water vapours and Tri-ethylene Glycol/water vapours in 2 m glass reactor

| Organics | Growth Factor (equivalent κ–value) | | | |
|---|---|---|---|---|
| | 50 nm | 75 nm | 100 nm | 150 nm |
| Propylene Glycol | 1.96 (0.87) | 2.11 (1.02) | 2.19 (1.11) | 2.41 (1.46) |
| Butylene Glycol | 1.78 (0.58) | 1.91 (0.81) | 1.98 (0.83) | 2.14 (1.08) |
| Tri-ethylene Glycol | 2.88 (2.73) | 3.09 (3.39) | 2.95 (2.94) | 2.73 (2.32) |




**Figures**



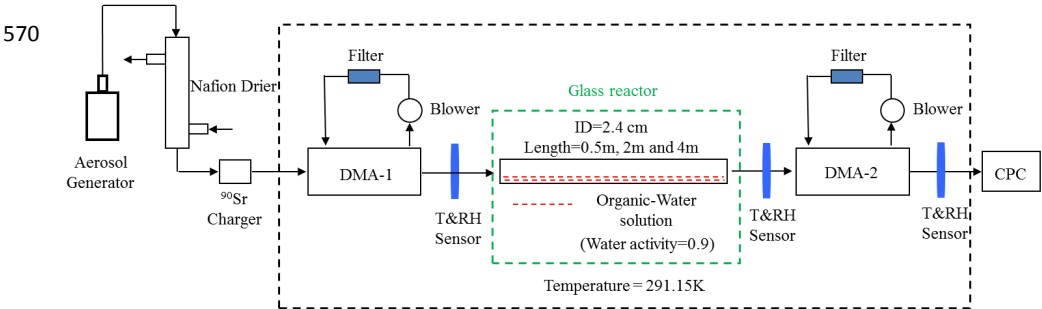

575              **Figure 1.** Schematic diagram of the experimental configuration





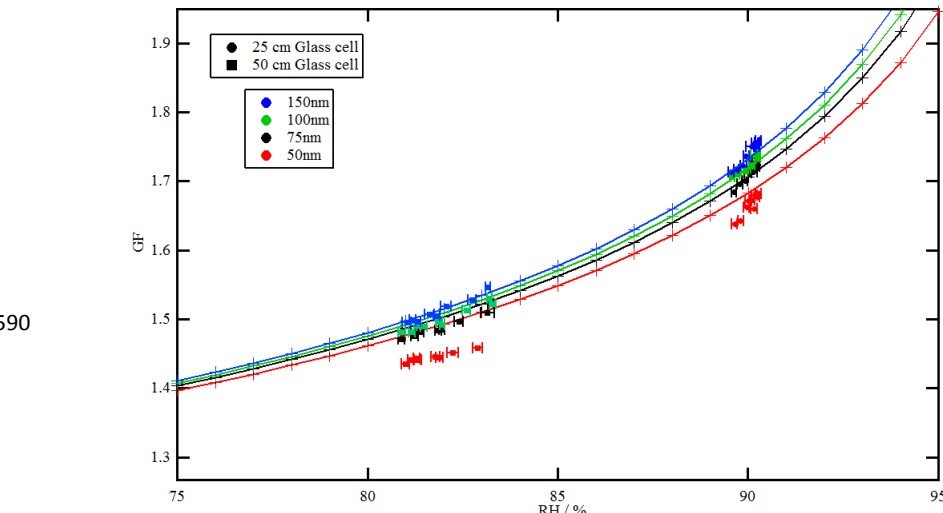

**Figure 2.** GF of $(NH_4)_2SO_4$ particles after transit through the 0.25 and 0.5 m glass reactor

containing $NaCl-H_2O$ solution with the water activity at 0.9.






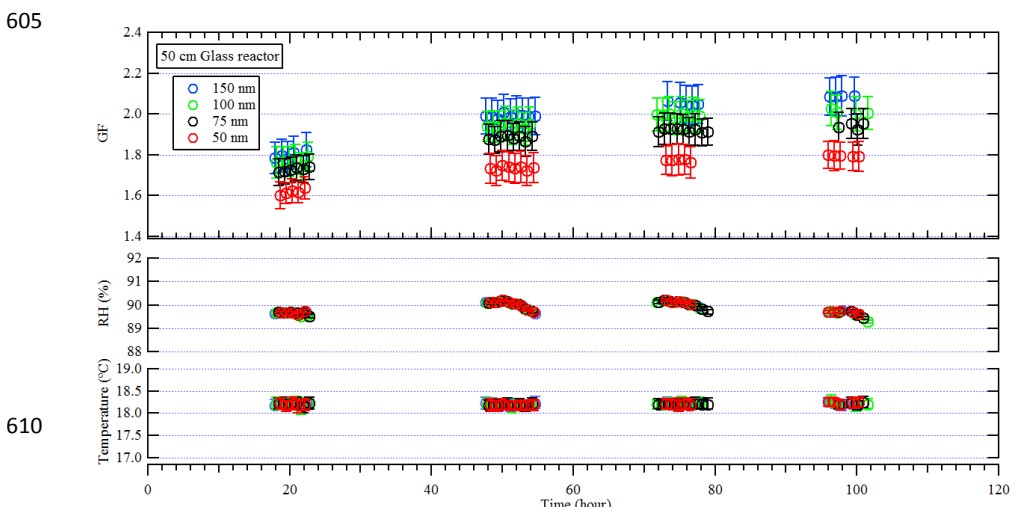

**Figure 3.** Time taken to achieve stable measurements in the HTDMA system. Steady state was only attained after several days, after which the GF remained constant for each organic system.





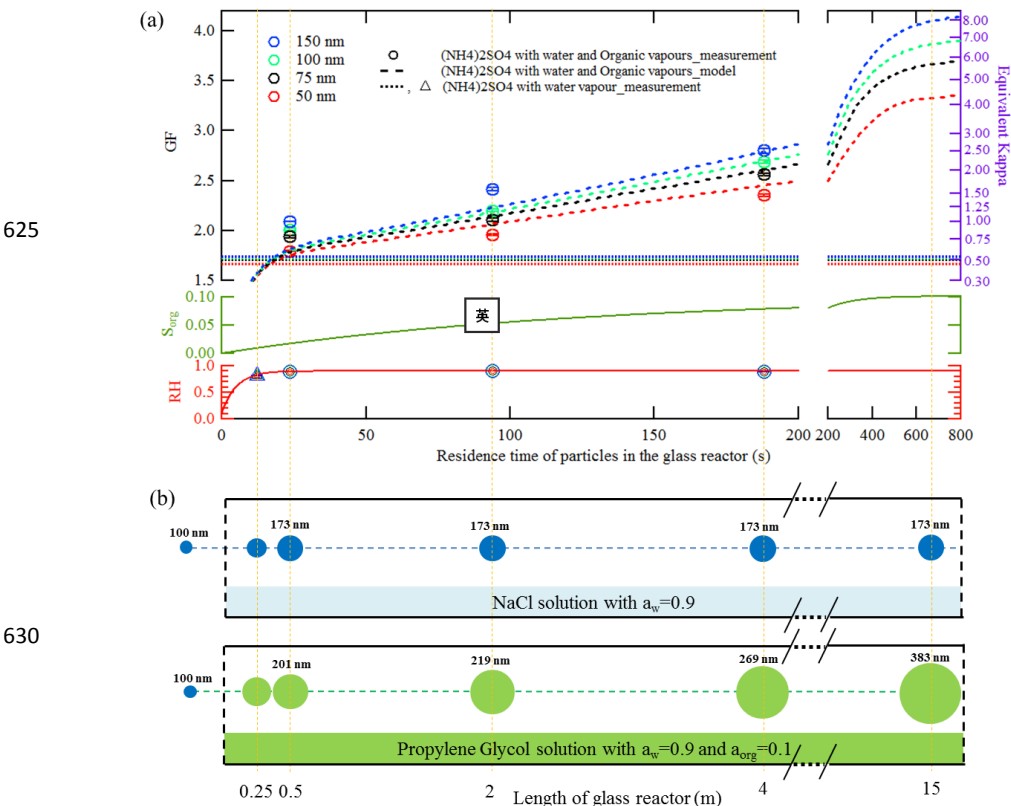

**Figure 4.** Growth of $(NH_4)_2SO_4$ particles exposed to water and propylene glycol, PG, vapours. (a) Measured (circle) and simulated (dash line) GFs with the residence time, the equivalent hygroscopicity (the $\kappa$ that an involatile particle would need in order to show the growth factor in the absence of co-condensing vapours) corresponding to the GF axis is presented on the right hand axis of the top panel, the simulated RH and $S_{org}$ profile are presented as solid red line and green line. (b) Schematic illustration of 100 nm $(NH_4)_2SO_4$ particles growth in the glass reactor, particle diameters to scale.




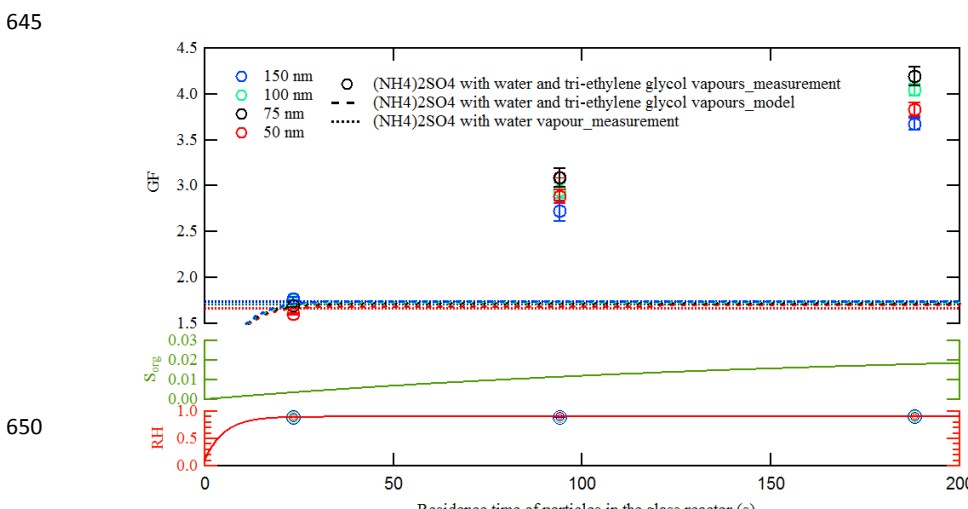

**Figure 5.** Measured (circle) and simulated GF (dash line) with residence time for $(NH_4)_2SO_4$

particles exposed to water and TEG vapours. The simulated RH and $S_{org}$ profile, which

following the same exponential function as the experiment with PG vapour, are presented as

solid red line and green line.







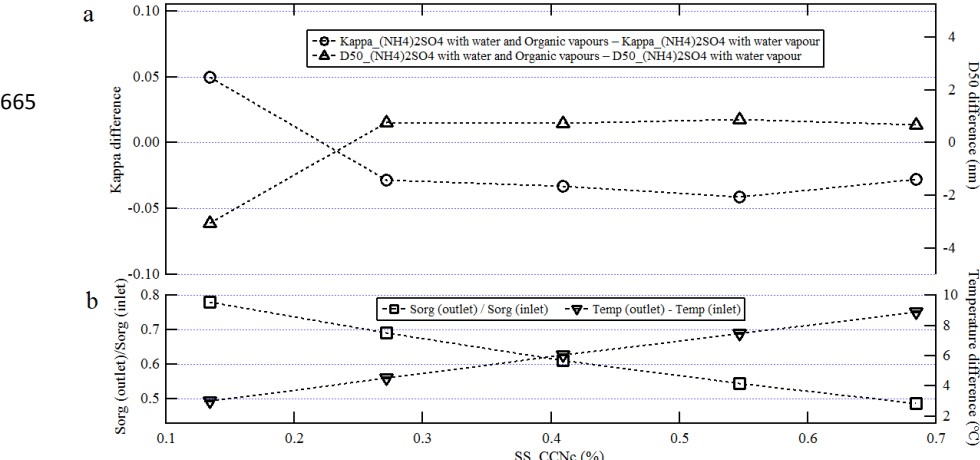

**Figure 6.** (a) Kappa and $D_{50}$ difference of $(NH_4)_2SO_4$ particles measured with and without

organic vapours in the sample and sheath air in the CCNc, (b) Temperature difference and the

corresponding calculated ratio of $S_{org}$ between outlet and inlet of the CCNc under different SS.





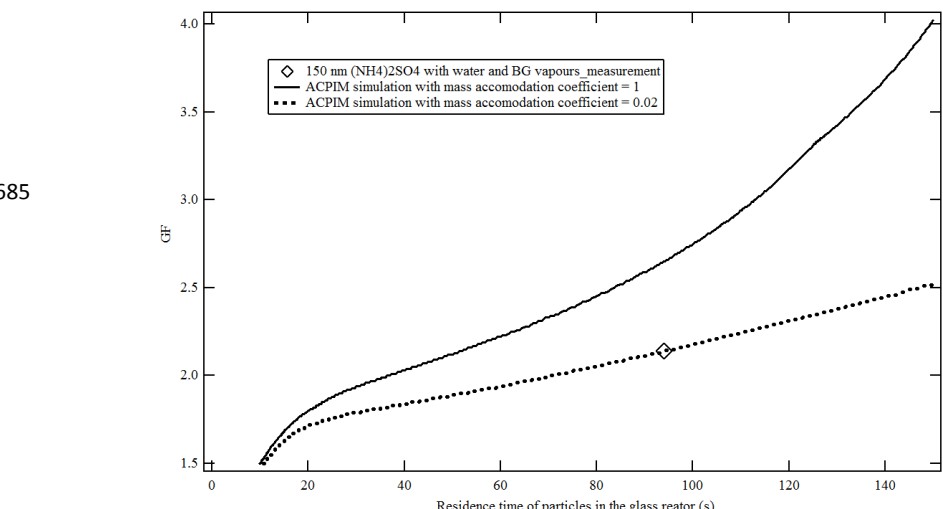

**Figure 7.** Measured and simulated GF with residence time for $(NH_4)_2SO_4$ particles exposed

to water and BG vapours.






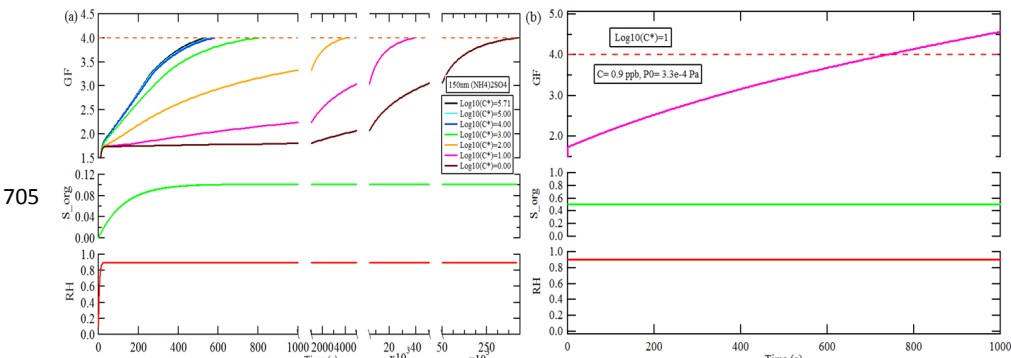



**Figure 8**. Simulated GF with residence time for: (a) organics with wide range of volatility, keep the same $S_{org}$ and RH profile as this study. (b) $\log_{10}(C^*)=1$ organic, the $S_{org}$ was increased to 0.5.






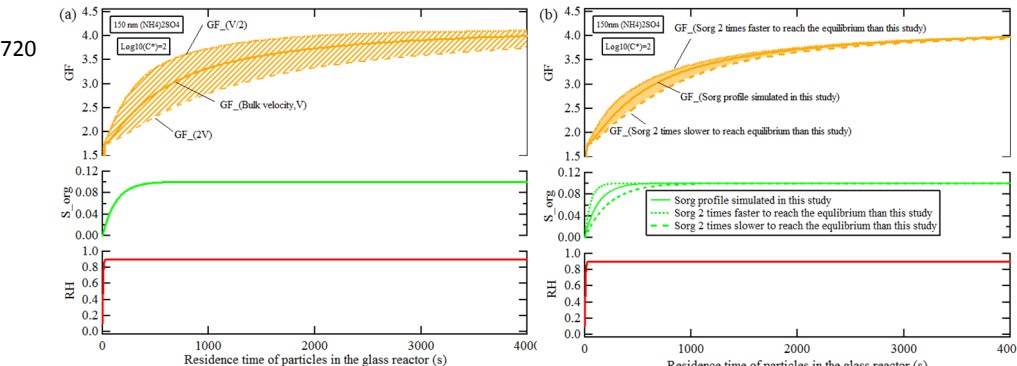

**Figure 9**. Simulated GF with residence time for organics with $\log_{10}(C^*)=2$: sensitivity to (a)

axial variability; (b) radial variability.