# Peer review of "Measured particle water uptake enhanced by co-condensing vapours"

_Atmospheric Chemistry and Physics, 2018_

## Referee Comment (RC1) · Anonymous Referee #1 · 11 Jul 2018

I have been reviewing this paper for Nature Communications and was supprised that it was not accepted. The authors have reformatted the text to a ACP style and resolved most of my concerns in this version.

This study investigated the role of co-condensation of organic vapors on the water uptake which has important implications for the aerosol-cloud interactions and their climate effects. The authors have cleverly designed their experiment and provided an experimental proof of the importance of co-condensation processes, which is an impossible mission for conventional HTDMA or CCN counter. Overall, it is an interesting and convincing study that may help improve our understanding on the cloud formation processes. Previous work was properly referred and the paper was clear and well written. I would strongly recommend its publication if the authors could address a couple

of minor issues listed below.

(1) More details needed for the model configuration: Fig. 1 shows that Sorg was not measured, so how was it calculated? How the evaporation of water vapor and organic vapor from the bath was treated in the model? How the loss on the glass reactor walls was considered? It seems that the sample flow has an initial RH of 0.0 but a non-zero Sorg on entry to the reactor (Fig. 1), should Sorg also be zero? Ln 292, "The Sorg profile was used to optimise the fit between simulated and measured GFs.", the authors should give more details, how you get the Sor profile and how to use it for the optimization.

(2) Thermodynamic or kinetic effect? Ln 294, " RH rapidly reaches equilibrium ($\sim$ 20 s) in the glass reactor, while Sorg needs longer ($\sim$ 700 s) (corresponding to 15 m glass reactor). This is expected since water is more volatile than propylene glycol, and water vapour lost to the walls will be more rapidly replenished from the solution than the propylene glycol vapours" Here again, how to know that Sorg needs longer time if there is no measurement of it? The authors attribute the difference to different volatilities. I'd rather think the difference is due to different kinetics (diffusion, accommodation coefficient, etc) because a difference in saturation pressure should not change the equilibrium time.

(3) Abstract, it appears as if a kappa of 2.64 is for atmospheric relevant conditions. I am not sure if the authors want to say that. I would suggest the authors either to change the formulation or justify it.

(4) Figure 4 panel a, what's the meaning of the symbol in the middle of the figure (Sorg panel)? .

Reference: Murphy, D. M. and Fahey, D. W.: Mathematical Treatment of the Wall Loss of a Trace Species in Denuder and Catalytic-Converter Tubes, Anal. Chem., 59, 2753–2759, 1987. Li, G., Su, H., Li, X., Kuhn, U., Meusel, H., Hoffmann, T., Ammann, M., Poeschl, U., Shao, M., and Cheng, Y.: Uptake of gaseous formaldehyde by soil

surfaces: a combination of adsorption/desorption equilibrium and chemical reactions, Atmospheric Chemistry and Physics, 16, 10299-10311, 10.5194/acp-16-10299-2016, 2016.

---

## Referee Comment (RC2) · A. Laaksonen (Referee) · 6 Sep 2018

This paper presents an experimental study of the effect of three different organic vapors on ammonium sulfate particles' equilibrium growth at varying relative humidities. As such, it relates to the co-condensation effect in cloud drop formation, whereby semivolatile organic or inorganic vapors add soluble mass to an aerosol population undergoing cloud drop activation, which can result in enhanced cloud drop number concentrations. I find this study a welcome addition to the literature, and have only a few comments, mostly relating to past work.

In the abstract, it is stated that the "... enhancement of particle water uptake through co-condensation constitutes the first direct measurement of this process..." Similarly,

in the end of Section 4 it is claimed that the authors have "observed for the first time that co-condensation of organic vapours can significantly promote water uptake of aerosol particles…" I don't think these statements are quite correct. Wagner and coworkers have published results of binary vapor condensation rates (both water-nitric acid and water-propanol) from which the co-condensation enhancement can be directly seen. See Rudolf et al., J. Aerosol Sci. 22, S51, 1991; Rudolf et al., J. Aerosol Sci. 32, 913, 2001.

CCN counter experiments and their explanation. I think the authors should refer to Romakkaniemi et al. (AMT 7, 1377, 2014) who studied the evaporation of ammonium nitrate and condensation of nitric acid inside the DMT CCN counter.

It is said on lines 345-346 that the absolute magnitude of co-condensation depends on the organic saturation ratio and not the absolute concentration. I think it should be clarified here that this refers to equilibrium growth. At cloud drop activation, the absolute concentration of the co-condensing species matters a lot.

---

## Author Comment (AC1) · 25 Sep 2018

Please find the revised manuscript in attachment!

Please also note the supplement to this comment:
https://www.atmos-chem-phys-discuss.net/acp-2018-586/acp-2018-586-AC1-supplement.pdf

---

## Author Comment (AC2) · 25 Sep 2018

RE: Atmos. Chem. Phys. Discuss., https://doi.org/10.5194/acp-2018-586

Dear Editor,

Please find our reply and revised manuscript in response to the comments of the 2 reviewers. We are most grateful for their constructive suggestions. The manuscript "Measured particle water uptake enhanced by co-condensing vapours", has been revised according to reviewers' suggestions.

The manuscript is a revised submission with new line and page numbers in the text, with all changes marked in red bold. We confirm that the submission of this revised version have been approved by all of the authors listed on this manuscript.

[Figure]

Yours Sincerely, Gordon McFiggans
* * *
The following is a point-to-point response to the reviewers' comments.

Anonymous Referee #1:

RC1: I have been reviewing this paper for Nature Communications and was supprised that it was not accepted. The authors have reformatted the text to a ACP style and resolved most of my concerns in this version.

RC1 answer: We thank the reviewer for their second review of this paper.

RC1: This study investigated the role of co-condensation of organic vapors on the water uptake which has important implications for the aerosol-cloud interactions and their climate effects. The authors have cleverly designed their experiment and provided an experimental proof of the importance of co-condensation processes, which is an impossible mission for conventional HTDMA or CCN counter.

RC1 answer: We thank the reviewer for recognising the substantial challenges that our experiments had to overcome and for understanding that clever design was required to access the effect.

RC1: Overall, it is an interesting and convincing study that may help improve our understanding on the cloud formation processes.

RC1 answer: Furthermore, we are extremely grateful to the reviewer for being convinced by our study.

RC1: Previous work was properly referred and the paper was clear and well written. I would strongly recommend its publication if the authors could address a couple of minor issues listed below.

RC1 answer: We thank the reviewer for their recommendation and note the minor

nature of the concerns raised. Nevertheless, we believe we can convincingly address these as outlined below.

RC1: (1) More details needed for the model configuration: Fig. 1 shows that Sorg was not measured, so how was it calculated? How the evaporation of water vapor and organic vapor from the bath was treated in the model? How the loss on the glass reactor walls was considered? It seems that the sample flow has an initial RH of 0.0 but a nonzero Sorg on entry to the reactor (Fig. 1), should Sorg also be zero? Ln 292, "The Sorg profile was used to optimise the fit between simulated and measured GFs.", the authors should give more details, how you get the Sorg profile and how to use it for the optimization.

RC1 answer: The reviewer raised the same question when he/she review this paper in "Nature communication" previously. Actually, as shown in Figure 4 on page 27, in this manuscript we have already included the data generated using the Sorg profile starting from zero at the entry to the flow tube and following an exponential function. This is used as the ACPIM input and the exponential rise is tuned to achieve the best fit between the modelled GFs and the experimental GFs.

In our system, the water and organic vapours in the headspace of the reactor was diluted by the sample air (dry and without organic vapour) and lost by the sink of droplet (condensing) and inner wall, but it will be replenished from the bath. For the water vapour, we measured the RH at the end of glass cell with different lengths (0.25, 0.5, 2 and 4 m), thereby empirically accounting for all effects (including wall loss) in constraining the RH profile across the glass cell in the ACPIM. For the organic, the tuned Sorg profile in the ACPIM, which shows the best fit between the modelled GFs and experimental GFs is admittedly somewhat illustrative. We cannot directly measure the VOC profile in the reactor, but can demonstrate that the measured GF at the end of three different length reactors is captured reasonably well by our co-condensation model and the relative magnitude of the growth of the different sized particles is similarly well-represented. There is little constraint for a more detailed consideration of the

flow conditions within our reactor, but we contend that our analytical approach provides sufficient insight to support our argument that co-condensation can play a substantial role in increasing the water uptake of ambient particulates.

RC1: (2) Thermodynamic or kinetic effect? Ln 294, " RH rapidly reaches equilibrium ($\sim$ 20 s) in the glass reactor, while Sorg needs longer ($\sim$ 700 s) (corresponding to 15 m glass reactor). This is expected since water is more volatile than propylene glycol, and water vapour lost to the walls will be more rapidly replenished from the solution than the propylene glycol vapours" Here again, how to know that Sorg needs longer time if there is no measurement of it? The authors attribute the difference to different volatilities. I'd rather think the difference is due to different kinetics (diffusion, accommodation coefficient, etc) because a difference in saturation pressure should not change the equilibrium time.

RC1 answer: The boundary between what should be considered a thermodynamic and a kinetic effect is very blurred when considering volatility. Volatility is essentially the tendency of a liquid to evaporate. So a higher volatility liquid shows a greater tendency to evaporate with the evaporation rate positively correlated with the vapour pressure (see, for example, Mackay et al, 2014). The log10(C*) of water and propylene glycol at 18 °C are 7.18 and 5.46, respectively. Thus, water vapour lost to the walls will be very much more rapidly replenished from the solution than the propylene glycol vapours. The thermodynamic properties of the components lead to a combined kinetic and thermodynamic effect since a higher volatility component will lead to a higher concentration in the vapour phase and hence more collisions as well as a higher rate of evaporation from the bulk.

References: Mackay, D. and van Wesenbeeck, I.. Correlation of chemical evaporation rate with vapor pressure. Environmental Science & Technology. 48: 10259-10263, 2014.

RC1: (3) Abstract, it appears as if a kappa of 2.64 is for atmospheric relevant conditions. I am not sure if the authors want to say that. I would suggest the authors either to change the formulation or justify it.

RC1 answer: We use equivalent hygroscopicity parameter (kappa) in this study, which we define as the ïĄń that an involatile particle would need in order to show the growth factor in the absence of co-condensing vapours. We are making no statement about whether this is under an atmospherically relevant condition – however, we would contend that it is not unreasonable to expect conditions where particles experience a summed saturation ratio of all organic components equivalent to 0.1. In this case, at RH = 90%, the particles would exhibit an effective kappa of 2.64. It would not be necessary to have such a high molecular concentration of the lower volatility atmospheric organic compounds as it takes in our experiments with the relatively volatile PG to attain a saturation ratio of 0.1. It can be envisaged that such conditions will arise in the presence of continual oxidation of biogenic emissions above a forest or of anthropogenic compounds downwind of heavily industrialised regions.

RC1: (4) Figure 4 panel a, what's the meaning of the symbol in the middle of the figure (Sorg panel)?

RC1 answer: We thank the reviewer for their observation and first, we would like to apologise for presenting the redundant symbol in the middle of the Figure 4. We had deleted it from the Figure 4 in the revised manuscript.

Reference: Murphy, D. M. and Fahey, D. W.: Mathematical Treatment of the Wall Loss of a Trace Species in Denuder and Catalytic-Converter Tubes, Anal. Chem., 59, 2753–2759, 1987. Li, G., Su, H., Li, X., Kuhn, U., Meusel, H., Hoffmann, T., Ammann, M., Poeschl, U., Shao, M., and Cheng, Y.: Uptake of gaseous formaldehyde by soil surfaces: a combination of adsorption/desorption equilibrium and chemical reactions, Atmospheric Chemistry and Physics, 16, 10299-10311, 10.5194/acp-16-10299-2016, 2016.

[Figure]

2018.

---

## Author Comment (AC3) · 25 Sep 2018

RE: Atmos. Chem. Phys. Discuss., https://doi.org/10.5194/acp-2018-586

Dear Editor,

Please find our reply and revised manuscript in response to the comments of the 2 reviewers. We are most grateful for their constructive suggestions. The manuscript "Measured particle water uptake enhanced by co-condensing vapours", has been revised according to reviewers' suggestions.

The manuscript is a revised submission with new line and page numbers in the text, with all changes marked in red bold. We confirm that the submission of this revised version have been approved by all of the authors listed on this manuscript.

[Figure]

Yours Sincerely, Gordon McFiggans
* * *
Referee #2 (A. Laaksonen):

RC2: This paper presents an experimental study of the effect of three different organic vapors on ammonium sulfate particles' equilibrium growth at varying relative humidities. As such, it relates to the co-condensation effect in cloud drop formation, whereby semivolatile organic or inorganic vapors add soluble mass to an aerosol population undergoing cloud drop activation, which can result in enhanced cloud drop number concentrations. I find this study a welcome addition to the literature, and have only a few comments, mostly relating to past work.

In the abstract, it is stated that the ". . . enhancement of particle water uptake through co-condensation constitutes the first direct measurement of this process. . ." Similarly, in the end of Section 4 it is claimed that the authors have "observed for the first time that co-condensation of organic vapours can significantly promote water uptake of aerosol particles. . ." I don't think these statements are quite correct. Wagner and coworkers have published results of binary vapor condensation rates (both water-nitric acid and water-propanol) from which the co-condensation enhancement can be directly seen. See Rudolf et al., J. Aerosol Sci. 22, S51, 1991; Rudolf et al., J. Aerosol Sci. 32, 913, 2001.

RC2 answer: We thank the reviewer for providing the additional literatures which related to our work. We have cited them in Line 60-61 on Page 3. We think the reviewer is right, and we have modified the related sentences.

Line 10-11 on Page 1, the sentence "Until now, there has been no direct observational evidence of this process" was modified to "Until now, there has been very few direct observational evidence of this process"

Line 28-30 on Page 2, the sentence "This enhancement of particle water uptake

through co-condensation of vapours constitutes the first direct measurement of this process, which. . ." was modified to "This enhancement of particle water uptake through co-condensation of vapours constitutes the direct measurement of this process, which. . ."

Line 59-61 on Page 3, the sentence "There has been no previous direct measurement evidence for this process in either inorganic or organic systems and existing instrumentation . . ." was modified to "There has been very less previous direct measurement evidence for this process in either inorganic (Rudolf et al., 2001) or organic systems (Rudolf et al., 1991) and existing instrumentation . . ."

Line 423-424 on Page 18, the sentence "This current study has observed for the first time that co-condensation of organic vapours can significantly promote water uptake of aerosol particles, . . ." was modified to "This current study has observed that co-condensation of organic vapours can significantly promote water uptake of aerosol particles, . . ."

RC2: CCN counter experiments and their explanation. I think the authors should refer to Romakkaniemi et al. (AMT 7, 1377, 2014) who studied the evaporation of ammonium nitrate and condensation of nitric acid inside the DMT CCN counter.

RC2 answer: We thank the reviewer for providing the additional literature which related to our work. We have cited it and add one sentence in Line 278-280 on Page 12. "the same result was also observed by Romakkaniemi et al. (2014) in their investigation of the evaporation of ammonium nitrate and condensation of nitric acid inside the DMT CCN counter"

RC2: It is said on lines 345-346 that the absolute magnitude of co-condensation depends on the organic saturation ratio and not the absolute concentration. I think it should be clarified here that this refers to equilibrium growth. At cloud drop activation, the absolute concentration of the co-condensing species matters a lot.

RC2 answer: We thank the reviewer for clarifying our statement. Yes, the reviewer is

right, we should refer this statement to the situation of equilibrium growth of droplet. Line 347-348 on Page 15, the sentence "The absolute magnitude of co-condensation depends on the saturation ratio, . . ." was modified to "During the equilibrium growth of droplet, the absolute magnitude of co-condensation depends on the saturation ratio, . . ."
* * *